# Synchronous Muscle Synergy Evaluation of Jaw Muscle Activities during Chewing at Different Speeds, a Preliminary Study

**DOI:** 10.3390/brainsci13091344

**Published:** 2023-09-19

**Authors:** Marzieh Allami Sanjani, Ehsan Tahami, Gelareh Veisi

**Affiliations:** 1Department of Biomedical Engineering, Mashhad Branch, Islamic Azad University, Mashhad, Iran 9187147578; allamimarzieh@gmail.com; 2Department of Computer Engineering, Mashhad Branch, Islamic Azad University, Mashhad, Iran 9177948564

**Keywords:** masticatory muscles, chewing, surface electromyography, muscle synergy

## Abstract

Human mastication is a complex and rhythmic biomechanical process regulated by the central nervous system (CNS). Muscle synergies are a group of motor primitives that the CNS may combine to simplify motor control in human movement. This study aimed to apply the non-negative matrix factorization approach to examine the coordination of the masticatory muscles on both sides during chewing. Ten healthy individuals were asked to chew gum at different speeds while their muscle activity was measured using surface electromyography of the right and left masseter and temporalis muscles. Regardless of the chewing speed, two main muscle synergies explained most of the muscle activity variation, accounting for over 98% of the changes in muscle patterns (variance accounted for >98%). The first synergy contained the chewing side masseter muscle information, and the second synergy provided information on bilateral temporalis muscles during the jaw closing. Furthermore, there was robust consistency and high degrees of similarity among the sets of muscle synergy information across different rate conditions and participants. These novel findings in healthy participants supported the hypothesis that all participants in various chewing speed conditions apply the same motor control strategies for chewing. Furthermore, these outcomes can be utilized to design rehabilitation approaches such as biofeedback therapy for mastication disorders.

## 1. Introduction

The human masticatory process is inherently complicated in its genesis and diverse in its expression. In general, neurosensory and neuromotor components control the mastication process. Furthermore, chewing, the first digestive step, requires extraordinary coordination among all the muscles [1]. The characteristics of mandibular movement, interocclusal force, and the ability to triturate food may indicate the neuromotor component of mastication. Mastication comprises the integrated function of the cervical and orofacial sensorimotor systems [2]. Sensory inputs of the orofacial and cervical tissues provide sophisticated bilateral control of the jaw and orofacial muscles to move, grind, and mix the food with saliva.

The masticatory system’s health, function, or malfunction may be correctly diagnosed with the help of measurements of mandibular motions and masticatory performance [3,4]. It has been challenging to quantify these factors to create clinical norms that accurately reflect masticatory health and function. Therefore, due to the complex chewing process, there is a real need to develop quantitative methods for evaluating a person’s capability to chew foods effectively.

According to numerous studies, the stomatognathic system may be hampered by age, gender, and some systemic conditions, including postural abnormalities, diabetes, osteoporosis, cardiovascular and respiratory disorders, dietary alterations, or stroke [5,6,7,8,9,10,11]. However, several studies, such as [10,12], focused on connecting the central nervous system (CNS) and the stomatognathic system. Therefore, the connection between chewing, the primary function of the stomatognathic system, and CNS function is significant in this context.

Since human mastication is a complex process that consists of two movements: clenching (mandible movement only in the sagittal plane) and grinding (providing a circular path in the frontal plane), more than 20 muscles are involved in the control of the human mastication process. As a diagnostic method to evaluate the condition of muscles and the nerve cells that regulate them (motor neurons), electromyography (EMG) was developed [13]. The motor neurons that trigger muscle contractions give electrical impulses to the electrodes. EMG has thus been employed as a diagnostic tool for identifying disorder that impacts the neurological and muscular systems and detecting injury’s origin and spatial location. Following, quantitative EMG can provide the possibility of monitoring muscle activity during mastication without interfering with natural chewing behavior [14,15,16,17,18] and also in diagnosing temporomandibular disorders to assess muscle function [19,20,21,22,23]. Furthermore, numerous attempts at developing EMG-based approaches related to mastication for masticatory rehabilitation robots [24,25,26], dental patient training [2], food texture assessment [27,28,29,30], and speech and swallowing therapy [7,31] were proposed.

Since chewing can be considered a rhythmical task with three phases: opening, closing, and occlusal phase [32], to perform these subtasks successfully and sequentially, complicated muscle activation patterns are required. It is determined that the nervous system handles muscle synergies or groups of muscles that function together rather than all muscles separately [33]. Since muscle synergy portrays a time-invariant activation combination across muscles activated by a time-varying coefficient, recorded muscle activation patterns can be reconstructed by the sum of muscle synergies. Therefore, the chewing movement strategy may be regulated by shared pattern-generating networks. Several studies have investigated that the central nervous system creates and coordinates neural control mechanisms for rhythmic tasks such as walking [1,33,34,35] or pedaling [36,37,38,39] through the flexible combinations of similar muscle synergies. However, to the authors’ knowledge, the jaw muscle synergies during rhythmical chewing have not been considered.

The objective of the present study was to scrutinize muscle synergies during chewing on the right and left sides. Furthermore, different chewing speeds in the normal range were performed, and the muscle synergies were extracted. Therefore, the novelty and importance of this study lie in several key aspects:By considering a comprehensive exploration of muscle synergies during chewing, the study acknowledges the significant individual differences in the masticatory system, such as variations in teeth health status, gum hardness, and an individual’s chewing side preference. This approach provides a better understanding of how muscle synergies adapt to such constraints and is paramount in comprehending the intricacies of the masticatory process.Examining muscle synergies across different chewing speeds aims to provide insights into the adaptability and flexibility of the masticatory system. This nuanced approach not only evaluates the stability of muscle synergies but also investigates how their activation patterns may change in response to variations in chewing speed. This dynamic perspective on muscle synergies during chewing is a unique aspect of the study and can provide a valuable contribution to the field.

Finally, the remaining sections of the paper are organized as follows: Section 2 investigates the proposed approach for muscle synergy extraction during chewing gum, followed by Section 3, which presents the numerical findings provided by the suggested approach. Section 4 discusses this study’s results, and Section 5 summarizes the paper.

## 2. Materials and Methods

### 2.1. Participants

The present pilot study was conducted at the Biomedical Engineering Laboratory, Islamic Azad University of Mashhad, Iran, where ten [40] adult humans (three males) were enrolled between 18 and 41 years (38.38 ± 12.48).

All participants with approximately natural dentitions greater than 24 were free from hypo- or hypersalivation while chewing gum, which causes additional muscle contractions [12,41]. Furthermore, the participants could provide the relative ability to chew gum with both sides according to the teeth’ health status and follow the chew rhythm required by the protocol.

The following exclusion criteria were factors that can affect masticatory performance: the presence of headache, high-level stress [42,43,44], depression [45], orofacial pain, participants with uncontrolled diabetes who reported bruxism, and dental pain. Moreover, participants who suffered from temporomandibular disorder (TMD) symptoms and were examined by an expert dentist were excluded.

The study was approved by the local Ethics Committee of the Mashhad Islamic Azad University of Medical Science (IR.IAU.MSHD.REC.1400.073), and all participants reviewed and signed an informed consent. The reporting of the study follows the STROBE guidelines, using the checklist for cross-sectional studies [46].

### 2.2. Muscle Synergy Analysis

The step-by-step procedure of muscle synergies analysis during chewing can be described as follows: EMG signal acquisition, preprocessing, chewing cycle detection, muscle synergy extraction, inter-subject and inter-speed variability. Figure 1 displays the framework of the proposed muscle synergy extraction analysis.

#### 2.2.1. Experimental Protocol and Data Acquisition

Although more than 20 muscles control the human mastication’s rhythmicity and coordination process, the masseter and temporalis are mainly employed for clenching [24]. Therefore, bipolar surface electrodes (Skintact Conductive Adhesive Electrodes, Leonhard Lang GmbH, Innsbruck, Austria) were placed on the right and left masseter and temporalis. The ground electrode was attached to the forehead. According to the recommendation [47] and using palpation, when participants clenched their teeth, the muscle locations were identified. The skin was rubbed with alcohol swabs to decline skin oiliness and impedance. A FlexComp Infiniti encoder (Thought Technology Ltd., Montreal, QC, Canada) measured the EMG signals with a sampling rate of 2400 Hz.

Participants were given a gum base pellet (30 mm, Biodent, Tehran, Iran) and asked to chew for approximately two minutes at the normal rate before the experiment began with the right/left side. Employing gum and detention of chewing to one side of the dental arch declined the extraneous sources of variation, such as differences in food textures and tongue movement changes related to food transport.

The experiment included two blocks with six different speeds (20, 30, 40, 50, 60, and 75 cycles/min), and an in-house program controlled the chewing rate. The chewing was accomplished with the right and left sides in the first and second blocks, respectively. The timeline of chewing in each block is illustrated in Figure 2. All volunteers sat comfortably in front of the monitor and were instructed to follow the chewing rates with video and audio cues representing each chewing cycle start.

#### 2.2.2. Data Analysis

##### Preprocessing

The EMG signals were bandpass filtered (fourth-order, zero-lag type I Chebyshev digital filter, bandwidth 12–450 Hz) to attenuate the DC baseline, low and high-frequency noise, and motion artifacts [48].

##### Detection of Chewing Cycle Onset

By referring to the masseter muscle of the side by which the gum was chewed, the onset of each chewing cycle at different speeds can be detected with the Teager–Kaiser energy operator (TKEO) [49,50]. For this purpose, the EMG signal was filtered by a bandpass fourth-order Butterworth filter with 30 to 300 Hz bandwidth [50]. After applying TSKO, the EMG signal was rectified and smoothed with a zero-lag, fourth-order low-pass filter with a 50 Hz cutoff frequency. The baseline mean (*µ*) and standard deviation (*σ*) were computed to determine the threshold (*T*) as follows:(1)T=μ+hσ
where *h* is a preset variable defining the level of the threshold. The threshold level was set to 20 since it was empirically found to be the most robust and introduced the smallest detection errors.

##### Muscle Synergy Extraction

EMG signals of each chewing cycle were rectified and smoothed with a low-pass filter with a cutoff frequency of 10 Hz to obtain EMG envelopes. The EMG envelopes were resampled at 240 Hz. The amplitude of the EMG envelopes for each muscle was normalized to the average peak value over all cycles. Afterward, non-negative matrix factorization (NMF) [33,51,52] was performed for consecutive chewing cycles to extract synchronous muscle synergies. The EMG matrix (*M*) was decomposed into spatial muscle weightings (*W*), which are referred to as the muscle synergies and their temporal activation patterns (*C*) by NNMF according to (2).
(2)M=W.C+E
where *E* is the residual error matrix.

The variance accounted for (VAF) [53] that examines the goodness of fit between the actual and reconstructed EMG by muscle synergies was utilized to find the optimal number of muscle synergies. The number of extracted muscle synergies is essential to the synchronous muscle synergy model. Therefore, the optimal number of synergies was defined as the compromise between model parsimony and reconstruction accuracy. Accordingly, VAF can be applied to select the optimal number of extracted synergies.

## 3. Results

According to the experimental protocol, EMG waveforms were recorded from left and right temporal and masseter. Examples of synchronous averaged EMG activity patterns for each muscle investigated at each speed by right, and left-side chewing are depicted in Figure 3.

The general effect of different speeds on the synchronous averaging of EMG activity was similar. However, there were subtle differences in the pattern and timing of activation. Despite the interval between each cycle at high speed being shorter, the temporalis and masseter muscles were active longer during each cycle with less average fluctuation.

Synchronous muscle synergies varying from one to four for all participants and speeds were extracted by the NMF approach. Two sets of muscle synergies as the appropriate number of muscle synergies (VAF (%) > 98%) for all participants at various chewing speeds by right and left side were identified (Figure 4). Therefore, two muscle synergies can produce initial EMG patterns for all speeds. Furthermore, it can be hypothesized the muscle activation patterns were a combination of two synchronous muscle synergies with a specific scaling coefficient.

Figure 5 illustrates two sets of muscle synergies for functional interpretation mined from the right and left temporalis and masseter muscles during chewing by the NMF method. It represented the basic features of the muscle synergies extracted at six chewing speeds from all participants. The synergy vector and scaling coefficient of the muscle synergies can explain the relative contribution of each muscle synergy to the overall muscle activity pattern during chewing.

According to Figure 5 and the synergy vectors and scaling coefficients analysis, some attributes were determined for each muscle synergy. The first synergy provided a strong activation pattern by greater than 50% of overall activation by the right and left masseter muscles during right and left-side chewing. However, the second synergy supported the activation of the muscle pattern during chewing by temporalis of both sides, specifically the muscles in agreement with the side of chewing.

The combination of the muscle synergies accurately reconstructed the activation patterns of four muscles. Figure 6 displays the instance of muscle waveform reconstruction by combining two groups of muscle synergies.

The quality of the EMG activation pattern reconstruction by combining two muscle synergies of all participants at each speed during right and left-side chewing is illustrated in Figure 7. Furthermore, the VAF of muscle synergies for each participant at all speeds is reported in Figure 8.

The similarity criterion was employed to explore the comparison of muscle synergies. The maximum normalized scalar product between synergy activation coefficients was described as the similarity between two groups of muscle synergies. Figure 9 displays the similarity of all participants and all speeds, known as inter-subject and inter-speed variability, respectively.

Regarding Figure 9, the similarity between all speeds was very high (0.99), mainly for the second synergy.

## 4. Discussion

The primary purpose of this work was to investigate the CNS coordination during unilateral chewing under different speeds. Consequently, the muscle synergies extracted from various participants and rates were considered. Two synchronous muscle synergies extracted through the NMF algorithm can highlight the attributes of the EMG patterns recorded from the right and left of the temporalis and masseter muscles during chewing.

Two muscle synergies accounted for the most variability in the EMG signals of mastication muscles during chewing gum. Although both temporalis and masseter muscles elevate the mandible and cause the jaw to close, the prominent muscle was the chewing side masseter muscle in the first synergy. This supports the earlier fundamental concept that the masseter is the predominant jaw muscle activity during chewing [54]. Moreover, the temporalis of both sides was more pronounced in the second synergy. The greater part of the activity of the bilateral temporalis muscles coincided with the chewing side masseter muscle activity. Since the activation of the non-chewing side masseter muscle was not significant in both synergies, this might be necessary for other motor activities like head motions or stabilizing the head while chewing rhythmically [54]. Consequently, the muscle synergies associated with each other’s mastication muscles were consistent with the kinesiology of these muscles.

As achieved in previous studies [36,37], the similarity of each set of muscle synergies across different speeds and participants was evaluated. Regarding the high similarity values provided for participants at different rates, it was concluded that each participant could apply the two similar functional muscle synergies despite differences in chewing performance or oral health status. Therefore, two muscle synergies selection was appropriate to describe the involved muscle activity. Accordingly, it can be hypothesized that each participant shared the same locomotor strategies and similar neural mechanisms during chewing motion.

Furthermore, there were shared pattern-generation networks for controlling the chewing motion according to the basic muscle synergic patterns held stable and consistent across a wide range of normal speeds. Timing shifts between muscle activation at different speeds can reflect the difference in the relative duration of the jaw closing. Moreover, high-speed chewing can generate more powerful and stable force and provide more comfort for participants because it produces a similar motion to normal chewing. Eventually, although it can be concluded that the rheological behavior of the food affects the jaw velocities [32], the findings denoted that these variations could not represent differences in the locomotor strategy for chewing.

As expected, there were some variations in patterns between participants or one participant’s right and left-side chewing. These may relate to differences in predominant chewing side, jaw geometry, teeth shape, oral status, and sensitivity to pain or food texture, such as rheological behavior, hardness, and adhesion, especially dentures. Furthermore, reducing the number of natural teeth, particularly posterior teeth, gradually diminishes chewing capacity and escalates challenges in mastication [55]. However, the results can be reproducible and stable over time. Furthermore, there was no significant mastication side preference impact on the temporalis and masseter muscle activation [56].

One major drawback in our current study that could introduce certain biases into the results was the instability in the chewing conditions, including variations in the rheological properties of the gum, gum sticking to dentures, or the emergence of subtle indications of learning in chewing patterns during different cycles for each speed. Although learning chewing patterns might lead to alterations between cycles and potentially bias the study’s outcomes, harnessing the training-related plasticity can be employed for rehabilitation treatments, particularly those reliant on biofeedback [57].

It is proposed that further investigation should be essayed in considering the phase shift as the temporal relationship between the bilateral masseter and temporalis muscles with other methods such as time-variant muscle synergy analysis [36] or coherency analysis in the frequency domain [54]. 

The outcomes from this research constitute a starting point for providing a robust and simple control strategy for rehabilitation robots. These assistive devices can realize maintenance and recovery/strengthening of oral motor functions such as smooth chewing and effective biting by increasing and decreasing jaw opening and closing muscle activity. Moreover, The heuristic value of this approach lies in the fact that these can be applied to evaluate the proposed therapy methods, such as visual or auditory biofeedback [58] for TMD and whiplash-associated disorders during critical chewing performance with a robust and objective indicator.

## 5. Conclusions

The coordinated features in the jaw muscle EMG signals during rhythmic chewing using NMF methods were investigated. Due to the different chewing speeds, two muscle synergies were extracted. The chewing side masseter muscle was predominant in the first synergy, and bilateral temporalis muscles were more active in the second synergy. A high degree of similarity between synergies of different speeds and participants may propose applying the same shared control strategy during chewing. These findings suggest that analysis of muscle synergies of jaw muscles may be practical to produce robust control commands for control of the EMG-based masticatory robots or quantitatively assess the rehabilitation training process for temporomandibular disorders.

## Figures and Tables

**Figure 1 brainsci-13-01344-f001:**
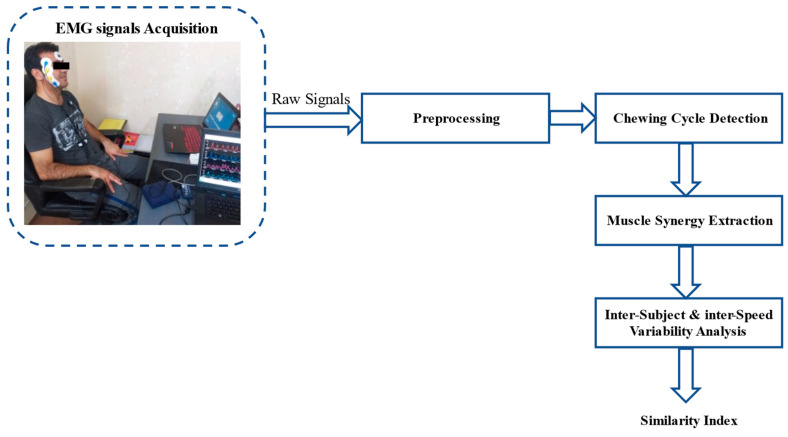
The framework of the proposed muscle synergy analysis.

**Figure 2 brainsci-13-01344-f002:**
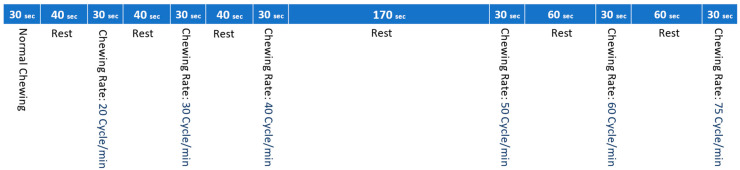
The timeline of chewing at different speeds.

**Figure 3 brainsci-13-01344-f003:**
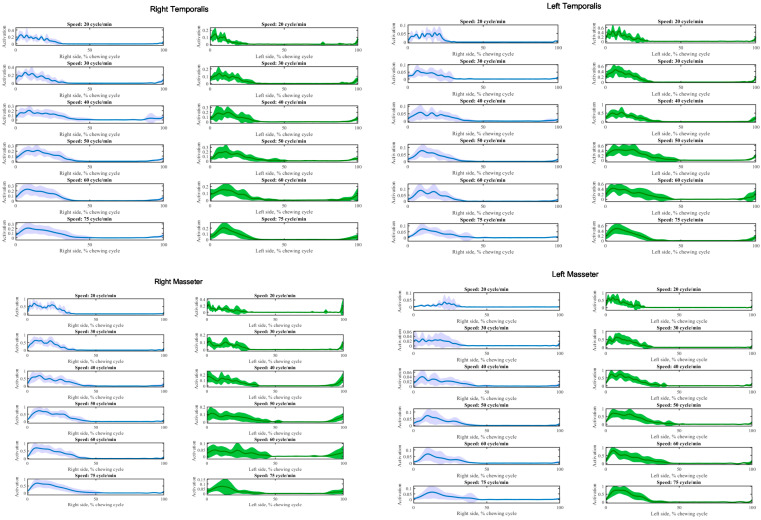
Synchronous averaging of temporalis and masseter muscle activation at each speed during chewing by right (blue) and left side (green) for one participant. The solid line and shaded area indicate the mean and standard deviation of muscle activation for all sequential chewing cycles. Each muscle activity was normalized by maximum activation among all chewing speeds.

**Figure 4 brainsci-13-01344-f004:**
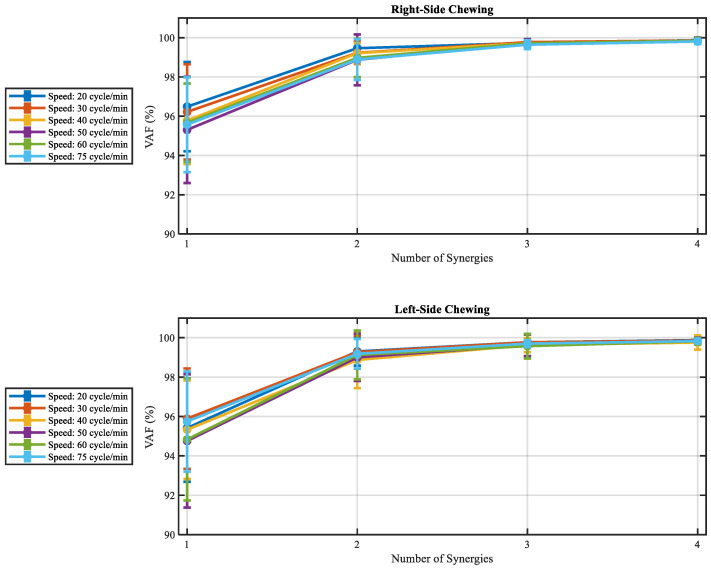
Mean and standard deviation across all participants and speed of VAF. Two synergies were for all participants and speeds.

**Figure 5 brainsci-13-01344-f005:**
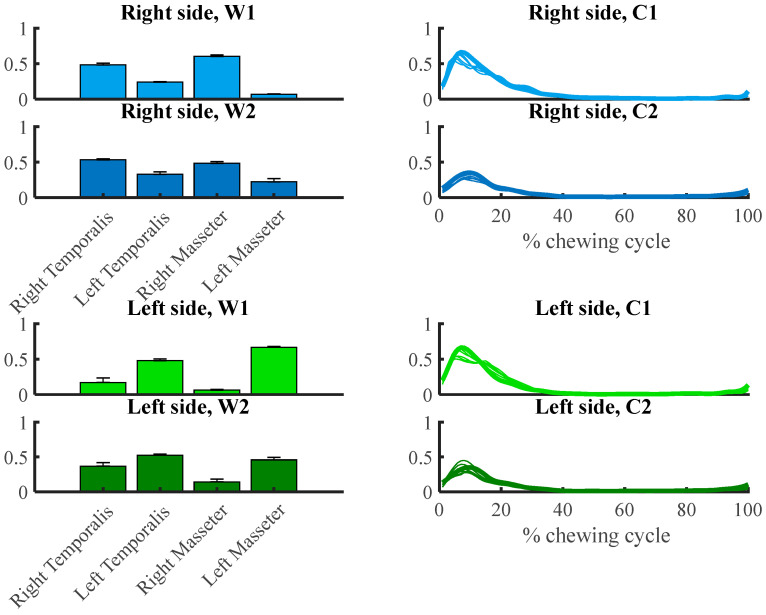
Muscle synergy vectors (W) and synergy activation coefficients (C) over all participants for two extracted synergy at different speeds. The error bar indicates the standard deviation of the muscle synergy vectors of all participants.

**Figure 6 brainsci-13-01344-f006:**
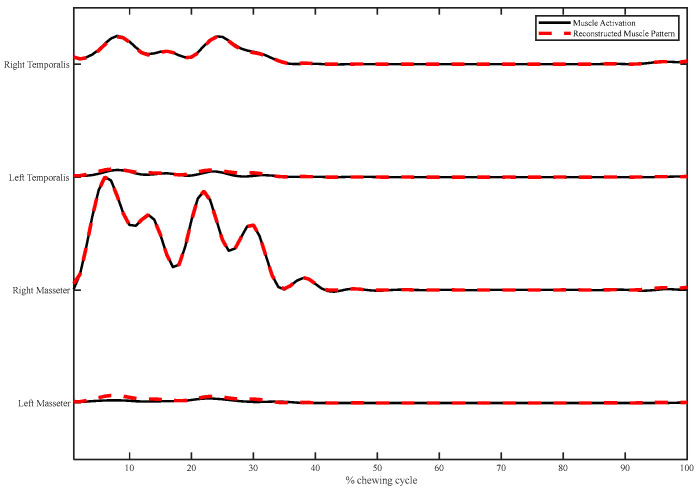
Individual illustration of the reconstructed muscle activation patterns (black solid line) at a speed of 50 cycles/min by combining two muscle synergies (red dotted line).

**Figure 7 brainsci-13-01344-f007:**
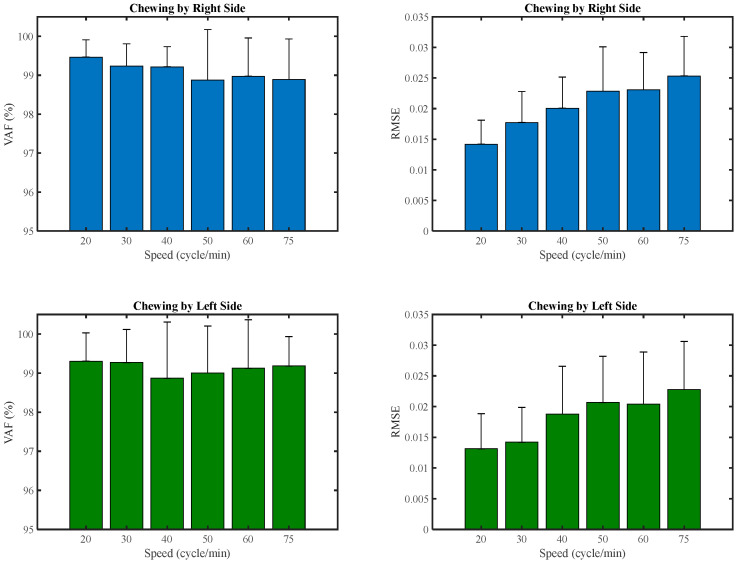
Mean and standard deviation of the VAF for all participants at each speed.

**Figure 8 brainsci-13-01344-f008:**
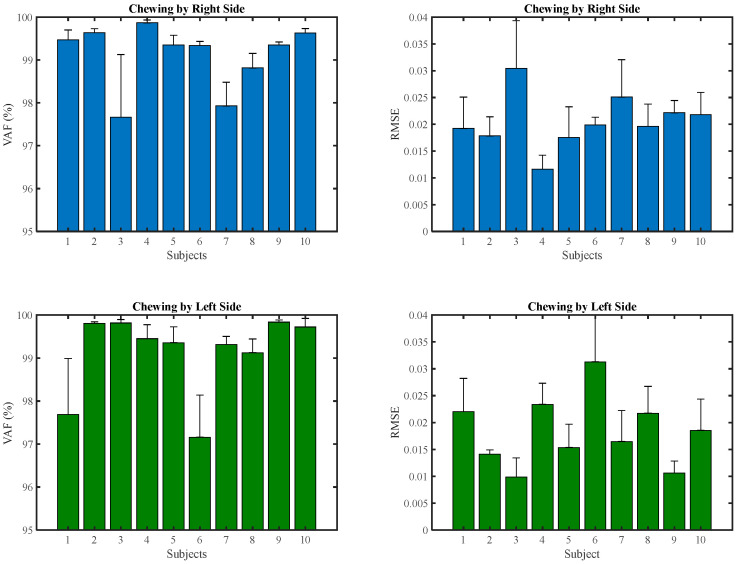
Mean and standard deviation of the VAF for each participant at each speed.

**Figure 9 brainsci-13-01344-f009:**
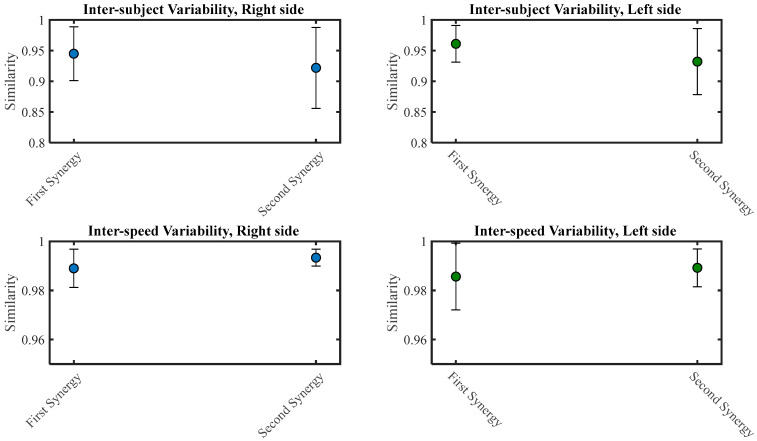
Error bar of inter-subject variability of similarity between two participants in each extracted synergy during right and left-side chewing (upper row). Error bar of similarity between two speeds (lower row).

## Data Availability

The data that support the findings of this study are available from the corresponding author.

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
