# Peer review of "Synchronous Muscle Synergy Evaluation of Jaw Muscle Activities during Chewing at Different Speeds, a Preliminary Study"

_brainsci, 2023, doi:10.3390/brainsci13091344_

Round 1
Reviewer 1 Report
The article seems to be a single Observational study as statistical analyses was absence
Moreover. methodological biases exist
(The Authors must see my remarks)

Author Response
Please see the attachment (Highlighted manuscript & point-by-point responses)
Comments from Reviewer #1
Comment 1:
Please clarify the type of the article, eg. research?
Response:
As per the journal format, we should label the research paper as "Article" in the following manner:
Type of the Paper (Article, Review, Communication, etc.)
Comment 2:
correct as "was"...
Response:
It was corrected.
Page 1, Line 14
During chewing, surface electromyography (sEMG) was recorded from the right and left masseter and temporalis muscles.
Comment 3:
References?
Response:
It was cited by the related reference.
Page 1, Line 29 & 30
Furthermore, chewing, the first digestive step, requires extraordinary coordination among all the muscles [1].
Comment 4:
such as?
Response:
It was updated as follows:
Page 2, Line 45 & 46
However, several studies, such as [2, 3], focused on connecting the central nervous system (CNS) and the stomatognathic system.
Comment 5:
references?
Response:
It was cited by the related reference.
Page 2, Line 52-54
As a diagnostic method to evaluate the condition of muscles and the nerve cells that regulate them (motor neurons), electromyography (EMG) was developed [4].
Comment 6:
Reduce that paragraph and state the aim of the article, clearly.....
Response:
We've made revisions to this paragraph to emphasize the study's unique contribution appropriately.
Page 2, Line 77-92
The objective of the present study was to scrutinize muscle synergies during chewing on the right and left sides. Furthermore, different chewing speeds in the normal range were performed, and the muscle synergies were extracted. Therefore, the novelty and importance of this study lie in several key aspects:
- By considering a comprehensive exploration of muscle synergies during chewing, the study acknowledges the significant individual differences in the masticatory system, such as variations in teeth health status, gum hardness, and an individual's chewing side preference. This approach provides a better understanding of how muscle synergies adapt to such constraints and is paramount in comprehending the intricacies of the masticatory process.
- By examining muscle synergies across different chewing speeds, the aim is to provide insights into the adaptability and flexibility of the masticatory system. This nuanced approach not only evaluates the stability of muscle synergies but also investigates how their activation patterns may change in response to variations in chewing speed. This dynamic perspective on muscle synergies during chewing is a unique aspect of the study and can provide a valuable contribution to the field.
Comment 7:
refrences?
Response:
In this section, we have introduced only the primary components of the analysis, deliberately avoiding the use of references. In the subsequent section, where the methods are introduced, we have incorporated relevant references where appropriate.
Comment 8:
How did the Authors determine the study size? Protocol? References? Inclusion / Exclusion Criteria?
Response:
We updated the Participant section to clarify the approach's rationale, enhancing the study's transparency and validity.
Page 3, Line 99-115
The present pilot study was conducted at the Biomedical Engineering Laboratory, Islamic Azad University of Mashhad, Iran, where ten [5] adult humans (three males) were enrolled between 18 and 41 years (38.38±12.48).
All participants with approximately natural dentitions greater than 24 were free from hypo- or hypersalivation while chewing gum, which causes additional muscle contractions [3, 6]. Furthermore, the participants could provide the relative ability to chew gum with both sides according to the teeth' health status and follow the chew rhythm required by the protocol.
The following exclusion criteria were factors that can affect masticatory performance: the presence of headache, high-level stress [7-9], depression [10], orofacial pain, participants with uncontrolled diabetes who reported bruxism, and dental pain. Moreover, participants who suffered from temporomandibular disorder (TMD) symptoms and were examined by an expert dentist were excluded.
The study was approved by the local Ethics Committee of the Mashhad Islamic Azad University of Medical Science (IR.IAU.MSHD.REC.1400.073), and all participants reviewed and signed an informed consent. The reporting of the study follows the STROBE guidelines, using the checklist for cross-sectional studies [11].
Comment 9:
References?
Response:
It was cited by the related references.
Page 3, Line 102-111
All participants with approximately natural dentitions greater than 24 were free from hypo- or hypersalivation while chewing gum, which causes additional muscle contractions [3, 6]. Furthermore, the participants could provide the relative ability to chew gum with both sides according to the teeth' health status and follow the chew rhythm required by the protocol.
The following exclusion criteria were factors that can affect masticatory performance: the presence of headache, high-level stress [7-9], depression [10], orofacial pain, participants with uncontrolled diabetes who reported bruxism, and dental pain. Moreover, participants who suffered from temporomandibular disorder (TMD) symptoms and were examined by an expert dentist were excluded.
Comment 10:
Do not state trademarks....
Response:
We've included the trademarks to ensure consistency in journal style and identify the tools and devices used.
Page 3, Line 125-126
… bipolar surface electrodes (Skintact Conductive Adhesive Electrodes, Leonhard Lang GmbH, Austria) …
Comment 11:
Do not state trademarks....
Response:
We've included the trademarks to ensure consistency in journal style and identify the tools and devices used.
Page 3, Line 130
A FlexComp Infiniti encoder (Thought Technology Ltd, Canada) …
Comment 12:
Do not state trademarks....
Response:
We've included the trademarks to ensure consistency in journal style and identify the tools and devices used.
Page 4, Line 132
… gum base pellet (30 mm, Biodent, Iran) …
Comment 13:
References?
Response:
Since the extent of initial gum softening relied on individual chewing habits and saliva production, each participant was instructed to chew the gum until it reached the desired softness for the experiment. Consequently, the duration of this process varied from person to person.
Page 4, Line 132-134
Participants were given a gum base pellet (30 mm, Biodent, Iran) and asked to chew for approximately two minutes at the normal rate before the experiment began with the right/left side.
Comment 14:
References?
Response:
The objective was to assess the variability in muscle synergy patterns during chewing under controlled conditions. Based on initial assessments of various participants and technical guidance from clinical experts, we selected speeds of 20, 30, 40, 50, 60, and 75 cycles/min for the study.
Page 4, Line 137-138
The experiment included two blocks with six different speeds (20, 30, 40, 50, 60, and 75 cycles/min), and an in-house program controlled the chewing rate.
Comment 15:
References?
Response:
We cited this statement as follows:
Page 4, Line 146-148
The EMG signals were bandpass filtered (fourth-order, zero-lag type I Chebyshev digital filter, bandwidth 12-450 Hz) to attenuate the DC baseline, low and high-frequency noise, and motion artifacts [12].
Comment 16:
References?
Response:
We cited this statement as follows:
Page 4, Line 152-153
For this purpose, the EMG signal was filtered by a bandpass fourth-order Butterworth filter with 30 to 300 Hz bandwidth [13].
Comment 17:
References?
Response:
When introducing the TKEO method, we cited the papers that have utilized the algorithm and its equations.
Page 4, Line 150-152
By referring to the masseter muscle of the side by which the gum was chewed, the onset of each chewing cycle at different speeds can be detected with the Teager–Kaiser energy operator (TKEO) [13, 14].
Comment 18:
References?
Response:
When introducing the NMF method, we cited the papers that have utilized the algorithm and its equations.
Page 4, Line 164-166
Afterward, non-negative matrix factorization (NMF) [15-17] was performed for consecutive chewing cycles to extract synchronous muscle synergies.
Comment 19:
According to that Section, the article seems to be an ''Observational'' one....,as ''p-value(s) are missing.....
Response:
In response to the insightful feedback from the esteemed referee, it's important to clarify that the test conducted in this study was of a pilot nature, primarily focused on gathering evidence. However, this research's principal aim was quantifying synergy-based information derived from EMG signals collected during each trial (chewing cycle) from individual participants during the chewing process. In practice, no comparisons between the observations were made to facilitate the application of hypothesis tests. However, the authors are open to revisiting the analysis and making necessary improvements if suggested to enhance the results' quality.
Comment 20:
Irrelevant data.....
Response:
The similarity analysis can provide valuable insights when assessing the resemblance of these synergies across various cycles within an individual or across different chewing speeds. In conclusion, the similarity analysis helped elucidate that the muscle synergy patterns, reflecting central nervous system behavior, remained remarkably consistent and were minimally affected by variations in conditions and EMG signal variability.
Comment 21:
How do we know that?
Response:
The reason for this statement, "Therefore, two muscle synergies selection was appropriate to describe the involved muscle activity. " was provided as follows:
Page 9, Line 259-262
Regarding the high similarity values provided for participants at different rates, it was concluded that each participant could apply the two similar functional muscle synergies despite differences in chewing performance or oral health status.
Comment 22:
Personal opinions or Conclusions?
Response:
According to scientific and qualitative observations, we could trace the relationship between jaw closing time and speed of chewing.
Page 9, Line 276-278
Timing shifts between muscle activation at different speeds can reflect the difference in the relative duration of the jaw closing.
Comment 23:
How do we know that?
Response:
Based on the initial examination of the participants' dental health by the clinical expert, as well as the questions posed during post-registration interviews, which included qualitative inquiries about the gum's hardness, its adherence to teeth or dentures, and tracking the signal quality during recording, several deductions were made.
Page 9, Line 276-278
These may relate to differences in predominant chewing side, jaw geometry, teeth shape, oral status, and sensitivity to pain or food texture, such as rheological behavior, hardness, and adhesion, especially dentures.
Comment 24:
Irrelevant conclusions.....
Response:
We wish to underscore that this study's primary focus was examining muscle synergies during chewing rather than conducting an extensive investigation into controlling an assistive robot for individuals with TMD. We intended to establish a foundational understanding of muscle synergy and coordination in healthy individuals. This knowledge can be necessary when designing control strategies for rehabilitation robots to produce the robust reference or desired trajectory.
Page 10, Line 294-295
The outcomes from this research constitute a starting point for providing a robust and simple control strategy for rehabilitation robots.
Comment 25:
Limitations? By the way, many methodological biases exist.....
Response:
We have included the study limitations in a separate paragraph, as follows:
Page 9, Line 283-289
One major drawback in our current study that could introduce certain biases into the results was the instability in the chewing conditions, including variations in the rheological properties of the gum, gum sticking to dentures, or the emergence of subtle indications of learning in chewing patterns during different cycles for each speed. Although learning chewing patterns might lead to alterations between cycles and potentially bias the study's outcomes, harnessing the training-related plasticity can be employed for rehabilitation treatments, particularly those reliant on biofeedback [18].
References
[1] A. Kumar et al., "Chewing and its influence on swallowing, gastrointestinal and nutrition-related factors: a systematic review," Critical Reviews in Food Science and Nutrition, pp. 1-31, 2022.
[2] S. S. Ulloa, A. L. C. Ordóñez, and V. E. B. Sardi, "Relationship between dental occlusion and brain activity: A narrative review," The Saudi Dental Journal, 2022.
[3] Y. Wu, Y. Lan, J. Mao, J. Shen, T. Kang, and Z. Xie, "The interaction between the nervous system and the stomatognathic system: from development to diseases," International Journal of Oral Science, vol. 15, no. 1, p. 34, 2023.
[4] M. Al-Ayyad, H. A. Owida, R. De Fazio, B. Al-Naami, and P. Visconti, "Electromyography Monitoring Systems in Rehabilitation: A Review of Clinical Applications, Wearable Devices and Signal Acquisition Methodologies," Electronics, vol. 12, no. 7, p. 1520, 2023.
[5] M. A. Hertzog, "Considerations in determining sample size for pilot studies," Research in nursing & health, vol. 31, no. 2, pp. 180-191, 2008.
[6] M. C. Verhoeff, M. Koutris, R. d. Vries, H. W. Berendse, K. D. v. Dijk, and F. Lobbezoo, "Salivation in Parkinson's disease: A scoping review," vol. 40, no. 1, pp. 26-38, 2023.
[7] T. P. S. Oei, S. Sawang, Y. W. Goh, and F. Mukhtar, "Using the depression anxiety stress scale 21 (DASS-21) across cultures," International Journal of Psychology, vol. 48, no. 6, pp. 1018-1029, 2013.
[8] K.-y. Kubo, M. Iinuma, and H. Chen, "Mastication as a stress-coping behavior," BioMed Research International, vol. 2015, 2015.
[9] Y. Tahara, K. Sakurai, and T. Ando, "Influence of chewing and clenching on salivary cortisol levels as an indicator of stress," Journal of Prosthodontics, vol. 16, no. 2, pp. 129-135, 2007.
[10] G. Zieliński et al., "Depression and Resting Masticatory Muscle Activity," Journal of Clinical Medicine, vol. 9, no. 4, p. 1097, 2020.
[11] E. Von Elm, D. G. Altman, M. Egger, S. J. Pocock, P. C. Gøtzsche, and J. P. Vandenbroucke, "The Strengthening the Reporting of Observational Studies in Epidemiology (STROBE) statement: guidelines for reporting observational studies," The Lancet, vol. 370, no. 9596, pp. 1453-1457, 2007.
[12] F. Davarinia and A. Maleki, "SSVEP-gated EMG-based decoding of elbow angle during goal-directed reaching movement," Biomedical Signal Processing and Control, vol. 71, p. 103222, 2022.
[13] S. Solnik, P. Rider, K. Steinweg, P. DeVita, and T. Hortobágyi, "Teager–Kaiser energy operator signal conditioning improves EMG onset detection," European journal of applied physiology, vol. 110, no. 3, pp. 489-498, 2010.
[14] X. Li, P. Zhou, and A. S. Aruin, "Teager–Kaiser energy operation of surface EMG improves muscle activity onset detection," Annals of biomedical engineering, vol. 35, no. 9, pp. 1532-1538, 2007.
[15] H. Yokoyama et al., "Basic locomotor muscle synergies used in land walking are finely tuned during underwater walking," Scientific Reports, vol. 11, no. 1, pp. 1-11, 2021.
[16] N. Dominici et al., "Locomotor primitives in newborn babies and their development," Science, vol. 334, no. 6058, pp. 997-999, 2011.
[17] M. C. Tresch, V. C. K. Cheung, and A. d'Avella, "Matrix factorization algorithms for the identification of muscle synergies: evaluation on simulated and experimental data sets," Journal of neurophysiology, vol. 95, no. 4, pp. 2199-2212, 2006.
[18] V. C. K. Cheung et al., "Plasticity of muscle synergies through fractionation and merging during development and training of human runners," Nature communications, vol. 11, no. 1, p. 4356, 2020.
[19] F. Hug, N. A. Turpin, A. Couturier, and S. Dorel, "Consistency of muscle synergies during pedaling across different mechanical constraints," Journal of neurophysiology, vol. 106, no. 1, pp. 91-103, 2011.
[20] S. Park and G. E. Caldwell, "Muscle synergies are modified with improved task performance in skill learning," Human Movement Science, vol. 83, p. 102946, 2022.
[21] B. Kibushi, S. Hagio, T. Moritani, and M. Kouzaki, "Speed-dependent modulation of muscle activity based on muscle synergies during treadmill walking," Frontiers in human neuroscience, vol. 12, p. 4, 2018.
[22] H. Yokoyama, T. Ogawa, N. Kawashima, M. Shinya, and K. Nakazawa, "Distinct sets of locomotor modules control the speed and modes of human locomotion," Scientific reports, vol. 6, no. 1, pp. 1-14, 2016.
[23] R. K. K. Ow, G. E. Carlsson, and S. Karlsson, "Relationship of masticatory mandibular movements to masticatory performance of dentate adults: a method study," Journal of oral rehabilitation, vol. 25, no. 11, pp. 821-829, 1998.
[24] M. Wieckiewicz, M. Zietek, D. Nowakowska, and W. Wieckiewicz, "Comparison of selected kinematic facebows applied to mandibular tracing," BioMed Research International, vol. 2014, 2014.
[25] A. Cuccia and C. Caradonna, "The relationship between the stomatognathic system and body posture," Clinics, vol. 64, no. 1, pp. 61-66, 2009.
[26] S. Nordio et al., "Biofeedback as an Adjunctive Treatment for Post-stroke Dysphagia: A Pilot-Randomized Controlled Trial," Dysphagia, vol. 37, no. 5, pp. 1207-1216, 2022/10/01 2022.
[27] Y. Ono, T. Yamamoto, K. y. Kubo, and M. Onozuka, "Occlusion and brain function: mastication as a prevention of cognitive dysfunction," Journal of oral rehabilitation, vol. 37, no. 8, pp. 624-640, 2010.
[28] F. B. Teixeira et al., "Masticatory deficiency as a risk factor for cognitive dysfunction," International journal of medical sciences, vol. 11, no. 2, p. 209, 2014.
[29] E. Ramazanoglu, B. Turhan, and S. Usgu, "Evaluation of the tone and viscoelastic properties of the masseter muscle in the supine position, and its relation to age and gender," Dental and Medical Problems, vol. 58, no. 2, pp. 155-161, 2021.
[30] W. Florjanski et al., "Evaluation of biofeedback usefulness in masticatory muscle activity management—A systematic review," Journal of clinical medicine, vol. 8, no. 6, p. 766, 2019.

Reviewer 2 Report
This is interesting study. However I found a few major and minor flaws:
1. The study included only ten participants which is the most important limitation of the study because of the risk of interpretation bias which is very high. Authors have to describe this limitation at the end of Discussion. The next limitation is a lack of control group. Furthermore Authors have to use the following title "Synchronous muscle synergy evaluation of jaw muscle activities during chewing at different speeds: a preliminary study".
2. Authors have to report the study strictly in accordance with the STROBE Statement for reporting observational studies. Please use the appropriate STROBE checklist for your type of study https://www.equator-network.org/reporting-guidelines/strobe/
3. Please describe detailed inclusion and exclusion criteria for study participants in Materials and Methods.
4. Authors wrote in Conclusions "These findings suggest that analysis of muscle synergies of jaw muscles may be practical to produce robust control commands for control of the EMG-based masticatory robots or quantitatively asses the rehabilitation training process for temporomandibular disorders.". Authors can't achieved such conclusion because they excluded participants with temporomandibular disorders. Please remove this sentence from conclusion.
5. Please describe in details how did you examine temporomandibular disorders? Which international recommendations did you use? DC/TMD or RDC/TMD or other?
6. Authors should cite this article Wieckiewicz M, Zietek M, Nowakowska D, Wieckiewicz W. Comparison of selected kinematic facebows applied to mandibular tracing. Biomed Res Int. 2014;2014:818694. doi: 10.1155/2014/818694 after this sentence "The masticatory system's health, function, or malfunction may be correctly diagnosed with the help of measurements of mandibular motions and masticatory performance [2]." because is highly reliable article which justify this sentence.
7. Please rationale the study and write why this study is novel and important within Introduction before the aim of the study.
8. Please write "According to numerous studies, the stomatognathic system may be hampered by age, gender, some systemic conditions, including postural abnormalities, diabetes, osteoporosis, cardiovascular and respiratory disorders, dietary alterations, or stroke [3-7]." instead of "According to numerous studies, the stomatognathic system may be hampered by some systemic conditions, including postural abnormalities, diabetes, osteoporosis, cardiovascular and respiratory disorders, dietary alterations, or stroke [3-7].". Authors should cite this article Ramazanoglu E, Turhan B, Usgu S. Evaluation of the tone and viscoelastic properties of the masseter muscle in the supine position, and its relation to age and gender. Dent Med Probl. 2021;58(2):155–161. doi:10.17219/dmp/132241 after the above mentioned sentence to justify this change.
9. Authors should discuss that EMG-biofeedback can be use to control masticatory muscle activity based on this study Florjanski W, Malysa A, Orzeszek S, Smardz J, Olchowy A, Paradowska-Stolarz A, Wieckiewicz M. Evaluation of Biofeedback Usefulness in Masticatory Muscle Activity Management-A Systematic Review. J Clin Med. 2019 May 30;8(6):766. doi: 10.3390/jcm8060766. This issue is strictly related to the reviewed study and should be discussed.
10. Authors have to provide ID of the Bioethical Committee approval.
The English Language is fine only minor corrections are required.
Author Response
Please see the attachment (Highlighted manuscript & point-by-point responses)
Comments from Reviewer #2
Comment 1:
The study included only ten participants which is the most important limitation of the study because of the risk of interpretation bias which is very high. Authors have to describe this limitation at the end of Discussion. The next limitation is a lack of control group. Furthermore Authors have to use the following title "Synchronous muscle synergy evaluation of jaw muscle activities during chewing at different speeds: a preliminary study".
Response:
We appreciate your insightful feedback, which has allowed us to clarify our rationale and approach, enhancing the transparency and validity of the study.
The decision to include only ten participants in the study is well-founded based on previous research on muscle synergy analysis in various conditions and movements [15, 19-22]. While the sample size may appear small, it's essential to understand that the data analysis approach compensates for this limitation. For each participant, multiple chewing cycles thorough analysis was conducted, encompassing a minimum of seven cycles per participant at the lowest speed (20 cycles/min). This resulted in a substantial dataset, with each participant providing more than 100 trials (chewing cycles). This extensive data collection ensures that the study included a suitable sample size, even within the context of a pilot study.
Furthermore, The absence of case and control groups is an intentional aspect of the study design, aligned with the research focus. Rather than comparing TMD individuals and a control group, we aim to comprehensively characterize muscle synergy patterns across a spectrum of conditions and variables for healthy individuals, such as different speeds, predominant chewing side, jaw geometry, teeth shape, or oral and teeth health status. This approach may provide the foundation for subsequent research, including investigations involving patient or intervention groups.
We agree that "Synchronous muscle synergy evaluation of jaw muscle activities during chewing at different speeds: a preliminary study" accurately represents the nature of the research. We will update the title accordingly in the revised manuscript.
Page 1, Line 2 & 3
Synchronous muscle synergy evaluation of jaw muscle activities during chewing at different speeds, a preliminary study
Comment 2:
Authors have to report the study strictly in accordance with the STROBE Statement for reporting observational studies. Please use the appropriate STROBE checklist for your type of study https://www.equator-network.org/reporting-guidelines/strobe/
Response:
Thank you for your valuable comment and for emphasizing the importance of adhering to reporting guidelines for observational studies like the STROBE Statement. Our study is indeed cross-sectional, and we were committed to reporting it strictly with the STROBE guidelines. Furthermore, We have included a statement in our manuscript to explicitly state our adherence to the STROBE guidelines, as follows:
Page 3, Line 118 & 119
The reporting of the study follows the STROBE guidelines, using the checklist for cross-sectional studies [11].
Comment 3:
Please describe detailed inclusion and exclusion criteria for study participants in Materials and Methods.
Response:
We have included detailed inclusion and exclusion criteria in the Materials and Methods section of the manuscript.
Page 3, Line 106-115
All participants with approximately natural dentitions greater than 24 were free from hypo- or hypersalivation while chewing gum, which causes additional muscle contractions [3, 6]. Furthermore, the participants could provide the relative ability to chew gum with both sides according to the teeth' health status and follow the chew rhythm required by the protocol.
The following exclusion criteria were factors that can affect masticatory performance: the presence of headache, high-level stress [7-9], depression [10], orofacial pain, participants with uncontrolled diabetes who reported bruxism, and dental pain. Moreover, participants who suffered from temporomandibular disorder (TMD) symptoms and were examined by an expert dentist were excluded.
Comment 4:
Authors wrote in Conclusions "These findings suggest that analysis of muscle synergies of jaw muscles may be practical to produce robust control commands for control of the EMG-based masticatory robots or quantitatively asses the rehabilitation training process for temporomandibular disorders.". Authors can't achieved such conclusion because they excluded participants with temporomandibular disorders. Please remove this sentence from conclusion.
Response:
We appreciate your careful review of our manuscript and the valuable feedback you have provided. We understand your concern regarding the statement in the Conclusions section of our paper, which suggests that our findings may have implications for the control of EMG-based masticatory robots and the assessment of rehabilitation training for temporomandibular disorders (TMD) despite excluding participants with TMD from our study.
We want to clarify our intention behind this statement. While it is true that we excluded participants with TMD from our current study, our research aimed to investigate muscle synergies during chewing and to establish a foundation for further investigations. We acknowledge that our study population may not directly represent individuals with TMD, and this limitation is duly noted. However, the study's findings can still provide insights into the broader understanding of muscle synergies according to the variability, even in healthy individuals.
Our statement in the Conclusions section was meant to highlight the potential future applications and directions for research in this field. By uncovering fundamental principles of muscle coordination in healthy individuals, we believe this study lays the groundwork for future studies that can specifically address muscle synergies in individuals with TMD. These future studies may build upon our findings to explore more practical applications, such as the control of EMG-based masticatory robots or the quantitative assessment of rehabilitation training for TMD.
In summary, while we acknowledge the limitations of our current study, we see it as a stepping stone for further research that can directly address the concerns raised in your comment.
Comment 5:
Please describe in details how did you examine temporomandibular disorders? Which international recommendations did you use? DC/TMD or RDC/TMD or other?
Response:
We would like to emphasize that this study focused on examining muscle synergies during chewing and did not aim to provide a detailed investigation of TMD itself. Instead, we intended to establish a baseline understanding of muscle synergy and coordination in healthy individuals, which can serve as a reference for future studies that may specifically address TMD-related issues.
In conclusion, we utilized the guidelines of an expert dentist, which encompassed evaluating jaw function, range of motion, and any signs of TMD-related pathology. We appreciate your feedback and have included this information in the revised manuscript to enhance the transparency of the methodology.
Page 3, Line 113-115
Moreover, participants who suffered from temporomandibular disorder (TMD) symptoms and were examined by an expert dentist were excluded.
Comment 6:
Authors should cite this article Wieckiewicz M, Zietek M, Nowakowska D, Wieckiewicz W. Comparison of selected kinematic facebows applied to mandibular tracing. Biomed Res Int. 2014;2014:818694. doi: 10.1155/2014/818694 after this sentence "The masticatory system's health, function, or malfunction may be correctly diagnosed with the help of measurements of mandibular motions and masticatory performance." because is highly reliable article which justify this sentence.
Response:
We incorporated this reference into the manuscript to ensure the readers can access this research. This will enhance the credibility and validity of the statement regarding diagnosing the masticatory system's health and function as a reliable source.
Page 1, Line 36-38
The masticatory system's health, function, or malfunction may be correctly diagnosed with the help of measurements of mandibular motions and masticatory performance [23, 24].
Comment 7:
Please rationale the study and write why this study is novel and important within Introduction before the aim of the study.
Response:
We appreciate your suggestion to provide a more explicit rationale for the study and highlight its novelty and importance within the Introduction section. We have revised the Introduction accordingly:
Page 2, Line 77-92
The objective of the present study was to scrutinize muscle synergies during chewing on the right and left sides. Furthermore, different chewing speeds in the normal range were performed, and the muscle synergies were extracted. Therefore, the novelty and importance of this study lie in several key aspects:
- By considering a comprehensive exploration of muscle synergies during chewing, the study acknowledges the significant individual differences in the masticatory system, such as variations in teeth health status, gum hardness, and an individual's chewing side preference. This approach provides a better understanding of how muscle synergies adapt to such constraints and is paramount in comprehending the intricacies of the masticatory process.
- By examining muscle synergies across different chewing speeds, the aim is to provide insights into the adaptability and flexibility of the masticatory system. This nuanced approach not only evaluates the stability of muscle synergies but also investigates how their activation patterns may change in response to variations in chewing speed. This dynamic perspective on muscle synergies during chewing is a unique aspect of the study and can provide a valuable contribution to the field.
Comment 8:
Please write "According to numerous studies, the stomatognathic system may be hampered by age, gender, some systemic conditions, including postural abnormalities, diabetes, osteoporosis, cardiovascular and respiratory disorders, dietary alterations, or stroke [3-7]." instead of "According to numerous studies, the stomatognathic system may be hampered by some systemic conditions, including postural abnormalities, diabetes, osteoporosis, cardiovascular and respiratory disorders, dietary alterations, or stroke [3-7].". Authors should cite this article Ramazanoglu E, Turhan B, Usgu S. Evaluation of the tone and viscoelastic properties of the masseter muscle in the supine position, and its relation to age and gender. Dent Med Probl. 2021;58(2):155–161. doi:10.17219/dmp/132241 after the above mentioned sentence to justify this change.
Response:
We have updated the sentence as you recommended:
Page 1, Line 42-45
According to numerous studies, the stomatognathic system may be hampered by age, gender, and some systemic conditions, including postural abnormalities, diabetes, osteoporosis, cardiovascular and respiratory disorders, dietary alterations, or stroke [1, 2, 25-29].
Comment 9:
Authors should discuss that EMG-biofeedback can be use to control masticatory muscle activity based on this study Florjanski W, Malysa A, Orzeszek S, Smardz J, Olchowy A, Paradowska-Stolarz A, Wieckiewicz M. Evaluation of Biofeedback Usefulness in Masticatory Muscle Activity Management-A Systematic Review. J Clin Med. 2019 May 30;8(6):766. doi: 10.3390/jcm8060766. This issue is strictly related to the reviewed study and should be discussed.
Response:
Given that the EMG signal stands as a pivotal instrument for appraising the efficacy of biofeedback treatment techniques [30], the findings from this study, centered on the extraction of muscle synergies during chewing, have the potential to serve as a robust and objective measure for assessing the outcomes of biofeedback treatment.
Page 10, Line 298-301
Moreover, The heuristic value of this approach lies in the fact that these can be applied to evaluate the proposed therapy methods, such as visual or auditory biofeedback [30] for TMD and whiplash-associated disorders during critical chewing performance with a robust and objective indicator.
Comment 10:
Authors have to provide ID of the Bioethical Committee approval.
Response:
We updated the Participants section with ID approval of the Ethics Committee as follows:
Page 3, Line 112-114
The study was approved by the local Ethics Committee of the Mashhad Islamic Azad University of Medical Science (IR.IAU.MSHD.REC.1400.073), and all participants reviewed and signed an informed consent.
References
[1] A. Kumar et al., "Chewing and its influence on swallowing, gastrointestinal and nutrition-related factors: a systematic review," Critical Reviews in Food Science and Nutrition, pp. 1-31, 2022.
[2] S. S. Ulloa, A. L. C. Ordóñez, and V. E. B. Sardi, "Relationship between dental occlusion and brain activity: A narrative review," The Saudi Dental Journal, 2022.
[3] Y. Wu, Y. Lan, J. Mao, J. Shen, T. Kang, and Z. Xie, "The interaction between the nervous system and the stomatognathic system: from development to diseases," International Journal of Oral Science, vol. 15, no. 1, p. 34, 2023.
[4] M. Al-Ayyad, H. A. Owida, R. De Fazio, B. Al-Naami, and P. Visconti, "Electromyography Monitoring Systems in Rehabilitation: A Review of Clinical Applications, Wearable Devices and Signal Acquisition Methodologies," Electronics, vol. 12, no. 7, p. 1520, 2023.
[5] M. A. Hertzog, "Considerations in determining sample size for pilot studies," Research in nursing & health, vol. 31, no. 2, pp. 180-191, 2008.
[6] M. C. Verhoeff, M. Koutris, R. d. Vries, H. W. Berendse, K. D. v. Dijk, and F. Lobbezoo, "Salivation in Parkinson's disease: A scoping review," vol. 40, no. 1, pp. 26-38, 2023.
[7] T. P. S. Oei, S. Sawang, Y. W. Goh, and F. Mukhtar, "Using the depression anxiety stress scale 21 (DASS-21) across cultures," International Journal of Psychology, vol. 48, no. 6, pp. 1018-1029, 2013.
[8] K.-y. Kubo, M. Iinuma, and H. Chen, "Mastication as a stress-coping behavior," BioMed Research International, vol. 2015, 2015.
[9] Y. Tahara, K. Sakurai, and T. Ando, "Influence of chewing and clenching on salivary cortisol levels as an indicator of stress," Journal of Prosthodontics, vol. 16, no. 2, pp. 129-135, 2007.
[10] G. Zieliński et al., "Depression and Resting Masticatory Muscle Activity," Journal of Clinical Medicine, vol. 9, no. 4, p. 1097, 2020.
[11] E. Von Elm, D. G. Altman, M. Egger, S. J. Pocock, P. C. Gøtzsche, and J. P. Vandenbroucke, "The Strengthening the Reporting of Observational Studies in Epidemiology (STROBE) statement: guidelines for reporting observational studies," The Lancet, vol. 370, no. 9596, pp. 1453-1457, 2007.
[12] F. Davarinia and A. Maleki, "SSVEP-gated EMG-based decoding of elbow angle during goal-directed reaching movement," Biomedical Signal Processing and Control, vol. 71, p. 103222, 2022.
[13] S. Solnik, P. Rider, K. Steinweg, P. DeVita, and T. Hortobágyi, "Teager–Kaiser energy operator signal conditioning improves EMG onset detection," European journal of applied physiology, vol. 110, no. 3, pp. 489-498, 2010.
[14] X. Li, P. Zhou, and A. S. Aruin, "Teager–Kaiser energy operation of surface EMG improves muscle activity onset detection," Annals of biomedical engineering, vol. 35, no. 9, pp. 1532-1538, 2007.
[15] H. Yokoyama et al., "Basic locomotor muscle synergies used in land walking are finely tuned during underwater walking," Scientific Reports, vol. 11, no. 1, pp. 1-11, 2021.
[16] N. Dominici et al., "Locomotor primitives in newborn babies and their development," Science, vol. 334, no. 6058, pp. 997-999, 2011.
[17] M. C. Tresch, V. C. K. Cheung, and A. d'Avella, "Matrix factorization algorithms for the identification of muscle synergies: evaluation on simulated and experimental data sets," Journal of neurophysiology, vol. 95, no. 4, pp. 2199-2212, 2006.
[18] V. C. K. Cheung et al., "Plasticity of muscle synergies through fractionation and merging during development and training of human runners," Nature communications, vol. 11, no. 1, p. 4356, 2020.
[19] F. Hug, N. A. Turpin, A. Couturier, and S. Dorel, "Consistency of muscle synergies during pedaling across different mechanical constraints," Journal of neurophysiology, vol. 106, no. 1, pp. 91-103, 2011.
[20] S. Park and G. E. Caldwell, "Muscle synergies are modified with improved task performance in skill learning," Human Movement Science, vol. 83, p. 102946, 2022.
[21] B. Kibushi, S. Hagio, T. Moritani, and M. Kouzaki, "Speed-dependent modulation of muscle activity based on muscle synergies during treadmill walking," Frontiers in human neuroscience, vol. 12, p. 4, 2018.
[22] H. Yokoyama, T. Ogawa, N. Kawashima, M. Shinya, and K. Nakazawa, "Distinct sets of locomotor modules control the speed and modes of human locomotion," Scientific reports, vol. 6, no. 1, pp. 1-14, 2016.
[23] R. K. K. Ow, G. E. Carlsson, and S. Karlsson, "Relationship of masticatory mandibular movements to masticatory performance of dentate adults: a method study," Journal of oral rehabilitation, vol. 25, no. 11, pp. 821-829, 1998.
[24] M. Wieckiewicz, M. Zietek, D. Nowakowska, and W. Wieckiewicz, "Comparison of selected kinematic facebows applied to mandibular tracing," BioMed Research International, vol. 2014, 2014.
[25] A. Cuccia and C. Caradonna, "The relationship between the stomatognathic system and body posture," Clinics, vol. 64, no. 1, pp. 61-66, 2009.
[26] S. Nordio et al., "Biofeedback as an Adjunctive Treatment for Post-stroke Dysphagia: A Pilot-Randomized Controlled Trial," Dysphagia, vol. 37, no. 5, pp. 1207-1216, 2022/10/01 2022.
[27] Y. Ono, T. Yamamoto, K. y. Kubo, and M. Onozuka, "Occlusion and brain function: mastication as a prevention of cognitive dysfunction," Journal of oral rehabilitation, vol. 37, no. 8, pp. 624-640, 2010.
[28] F. B. Teixeira et al., "Masticatory deficiency as a risk factor for cognitive dysfunction," International journal of medical sciences, vol. 11, no. 2, p. 209, 2014.
[29] E. Ramazanoglu, B. Turhan, and S. Usgu, "Evaluation of the tone and viscoelastic properties of the masseter muscle in the supine position, and its relation to age and gender," Dental and Medical Problems, vol. 58, no. 2, pp. 155-161, 2021.
[30] W. Florjanski et al., "Evaluation of biofeedback usefulness in masticatory muscle activity management—A systematic review," Journal of clinical medicine, vol. 8, no. 6, p. 766, 2019.

Reviewer 3 Report
Dears,
The paper has well-designed research methods, appropriate statistical analysis and a relatively good interpretation of the results.
Please be sure to use only keywords accordingly to medical subject headings (Mesh word) for a better indexing.
I suggest you add a table with the list of abbreviations used in the text.
I suggest you implement the abstract in order to make it more understandable to authors.
The introduction should be expanded perhaps by adding a section on temporomandibular disorders. I recommend some references:[10.1080/08869634.2016.1203560]
The conclusion is in accordance with the objectives of the research, its results and their interpretation, as well as the relevant literature.
Regards
There are some punctuation and syntax errors
Author Response
Please see the attachment.
Comments from Reviewer #3
Comment 1:
Please be sure to use only keywords accordingly to medical subject headings (Mesh word) for a better indexing.
Response:
We appreciate your suggestion, and We incorporated MeSH terms as extensively as possible in using keywords to improve the indexing.
Page 1, Line 24
Keywords: Masticatory Muscles; Chewing; Surface Electromyography; Muscle Synergy;
Comment 2:
I suggest you add a table with the list of abbreviations used in the text.
Response:
We've compiled the table of abbreviations in the following manner, but we've encountered difficulty in determining its proper placement within the journal template.
List of Abbreviations
|
Abbreviations |
Definitions |
|
CNS |
Central Nervous System |
|
EMG |
Electromyography |
|
TMD |
Temporomandibular Disorder |
|
TKEO |
Teager–Kaiser Energy Operator |
|
NMF |
Non-negative Matrix Factorization |
|
VAF |
Variance Accounted for |
Comment 3:
I suggest you implement the abstract in order to make it more understandable to authors.
Response:
Thank you for your suggestion on improving the clarity of the abstract, and we hope the enhancements will prove effective.
Page 1, Line 13-17
Ten healthy individuals were asked to chew gum at different speeds while their muscle activity was measured using surface electromyography of the right and left masseter and temporalis muscles. Regardless of the chewing speed, two main muscle synergies explained most of the muscle activity variation, accounting for over 98% of the changes in muscle patterns (variance accounted for >98%).
Comment 4:
The introduction should be expanded perhaps by adding a section on temporomandibular disorders. I recommend some references:[10.1080/08869634.2016.1203560]
Response:
Thank you for your valuable feedback. While we appreciate the suggestion to include a section on temporomandibular disorders in the introduction, it is essential to note that the primary focus and objective of our study is to scrutinize muscle synergies and coordination during chewing under variable conditions such as variations in teeth health status, gum hardness, an individual's chewing side preference, or different chewing speed. Therefore, we have intentionally limited the scope of the introduction to provide background information directly related to our study's objectives.

Round 2
Reviewer 2 Report
The manuscript has been correctly revised. I don't have further comments.